# Metal Accumulation Using a Bacterium (K-142) Identified from Environmental Microorganisms by the Screening of Au Nanoparticles Synthesis

**DOI:** 10.3390/ma13214922

**Published:** 2020-11-02

**Authors:** Yiting Li, Michio Suzuki

**Affiliations:** Department of Applied Biological Chemistry, Graduate School of Agricultural and Life Sciences, The University of Tokyo, 1-1-1 Yayoi, Bunkyo-ku, Tokyo 113-8657, Japan; liyiting1993@outlook.com

**Keywords:** *Bacillus*, metal, environmental microorganisms, screening, nanoparticles, accumulation, deposition

## Abstract

The use of technology that uses organisms to synthesize metal nanoparticles is necessary to maintain a sustainable society. In this study, we investigated and screened the microorganisms isolated from environmental water by quantifying the reproducibility of synthetic Au nanoparticles and the ability of large amount synthesis. The microorganism (K-142) of the *Bacillus* genus showed the best activity in the investigation. K-142 can also synthesize Ag, CdS and PbS nanoparticles, and the deposition efficiency of Ag, Al, Cd, Cu, and Pb was about 64.8–99.2%. According to the observation results under the microscope after fluorescent staining, K-142 could survive after being treated with 0.5 mM metal solution for 24 h. Therefore, it is expected that K-142, which is easy to cultivate, would also have a high ability to reduce and deposit metal substances. K-142 can be applied to the concentration and recovery of heavy metals in environmental water, thereby opening up channels for biological water purification.

## 1. Introduction

With the development of science and technology, the problems of environmental pollution caused by industrial waste have also emerged. Metal resources used in many fields are discharged in the form of industrial waste, which is often regarded as one of the main causes of environmental pollution. In order to reduce pollution, the recycling of heavy metal resources scattered in the environment has become one of the main priorities, through environmentally friendly methods. Utilizing living organisms in the environment is considered to be an environmentally friendly method for achieving this. The recycling method using microorganisms is considered to be most realistic method because of the costs and activities associated with it.

In contrast to bulk metals, metal nanoparticles with specific physical and chemical properties are considered to be materials with a broad application value and high development prospects [1]. The traditional method for synthesizing metal nanoparticles requires high cost conditions to be maintained such as high temperatures and high pressure. In addition, a large amount of organic solvents would also need to be used, which usually produces harmful by-products. This would impose a major burden on the environment. Therefore, the use of environmentally friendly methods involving microorganisms for the biosynthesis of metal nanoparticles has been attracting increasing attention in recent years. So far, synthesis of plate-shaped Fe_3_O_4_ nanoparticles by Magnetotactic bacterium MV-1 [2], synthesis of plate-shaped Au nanoparticles by *Pseudomonas*
*stutzeri* AG259 [3], synthesis of 2–5 nm CdS by *Escherichia*
*coli* [4], synthesis of ZnS nanoparticles extracellularly by using *Desulfobacter*
*aceae* [5], Au nanoparticles formation by *Rhodobacter*
*capsulatus* [6], biosynthesis of Au-Pd core-shell nanoparticles through *E*. *coli* [7], synthesis of Au nanoparticles by extracellular components of *Lactobacillus*
*casei* [8], ZnS nanolayers synthesized on the surface of *Phanerochaete*
*chrysosporium* [9], SiO_2_ nanoparticles synthesized by *Saccharomyces*
*cervisiae* [10], synthesis of CdSAg quantum dots and Ag_2_S nanoparticles through cation exchange of bacteria [11],and many other methods for the reduction of metal ions and formation of nanoparticles by bacterial cells have already been reported.

Our objective was to identify the microorganisms that have strong resistance against heavy metals and that accumulate heavy metals around the cell body. In this study, we focused on Au as the main heavy metal, and screened the microorganism K-142 (*Bacillus* genus), which is the best microorganism for synthesizing Au nanoparticles, from the microorganisms isolated from the environmental water obtained through sampling. When Au nanoparticles are synthesized, the solution turns red to purple due to surface plasmon resonance (SPR) [12]. We used the color change of Au nanoparticles synthesis for the screening of bacterial strains. This new concept of screening can identify the novel bacterial strains for bioremediation. Since this color change can be easily discerned by the eyes, we used the result of the color change as an indicator of the ability to synthesize Au nanoparticles to screen microorganisms. Subsequently, we conducted a metal accumulation test on the selected strain K-142. By measuring the abilities of this test to synthesis nanoparticles and accumulation efficiency of metals other than Au, we explored the possibility of its utilization in the field of metal recovery.

## 2. Materials and Methods

### 2.1. Collection and Culture of Microorganisms

We collected the samples of fresh environmental water from around the Kamioka mine in Hida City, Gifu Prefecture in Japan. The water samples containing solid sediments were collected at seven spots (K1 to K7), as shown in Appendix A (The map data of Appendix A is quoted from GSI Maps). Photos of each spot are shown in Appendix A.

In order to isolate the microorganisms in the collected water, we prepared a general-purpose medium called LB-Glucose (1% Meat Extract/1% Peptone/0.1% NaCl/1% Glucose/1.5% Agar), and used it to screen the microorganisms in the water samples. The compositions other than glucose were dissolved in distilled water and sterilized in an autoclave at 120 °C for 20 min. The glucose (aqueous solution) was separately sterilized (120 °C, 20 min) and added after the liquid was sufficiently cooled.

We diluted each of the collected environmental water samples and spread them on agar mediums. We then placed the mediums at 25 °C and cultured for 1 to 7 days. Afterwards, we observed the growth of the colony regularly during the cultivation process to avoid overproliferation. We then used the sterilized toothpicks to transplant the single colonies onto the agar mediums, which were divided into 24 grids, and placed the mediums at 25 °C. Through these steps, we isolated the microorganisms in the water samples. The process flow diagram is shown in Appendix A.

We created a library for the strains that were isolated. A total of 282 microorganisms were isolated in our study.

### 2.2. Screening by Adding Chloroauric Acid

In this study, we conducted two rounds of screenings by adding chloroauric acid (Sigma-Aldrich, St. Louis, MO, USA) (hereinafter referred to as “auric acid”). In the first round of screening, all 282 isolated strains were used as the target strains, and we conducted a rough selection based on the results after adding auric acid. Firstly, we added the liquid culture medium to the 24-well plate (1 mL to each well) in a clean bench. We then inoculated the single colony of each strain from the agar mediums with toothpicks. Afterwards, we added auric acid to the suspension (contained medium components) of each strain obtained by culturing for 24 h so we could have the same final concentration for each strain (1.0 mM). Then, the shaking culture process was carried out at 25 °C for 2 days. After that, we observed the color changes of the solution and took photos to record the results of each stage. Taking the strains in all the wells as the range, we selected several strains that turned from red to purple or had a large degree of color change according to our observation with the naked eye. In addition, the sampling spots and numbers corresponding to the selected strains were given as names and summarized in the form of a table.

The strains selected in the first round were used as the target strains in the second round. The evaluation was performed in stages as follows.

#### 2.2.1. Evaluation Based on the Coloration after Washing Out the Medium (Small Amount Synthesis)

Firstly, 1 mL of liquid culture was performed in 24-well plates (25 °C, 24 h). We centrifuged (4000× *g*, 4 °C, 10 min) the obtained cell suspension to remove the medium component in the supernatant. We then repeated this washing operation three times. We resuspended the washed pellets (bacterial cells) in 1 mL of sterilized water, and then added the auric acid. After we added auric acid, we let the water react for 48 h while shaking the mixture. After 48 h, the change in color of each well was confirmed and evaluated based on our observations. The evaluation was performed by digitizing the degree of the color change. The evaluation criteria are shown in Appendix A. The stains which changed to dark-purple or other dark-colored stains were calculated as 1 a.u., while the light-migrated or pale-purple strains were calculated as 0.5 a.u., and the strains with no color change were calculated as 0 a.u. The reproduction experiments were performed 7 times, with the examples shown in Appendix A.

We calculated the values of the 7 reproduction experiments in the form of a table. We then selected the strains with a total value greater than 4.5 a.u. We took them as the test subjects for the large-scale (50 mL) synthesis and continued to evaluate their synthesis ability.

#### 2.2.2. Evaluation Based on the Coloration after Washing Out the Medium (Large Amount Synthesis)

Strains with a total value of 4.5 a.u. or more were cultured in a 300 mL Erlenmeyer flask (50 mL, 25 °C, 24 h). After washing the medium components by centrifugation, we added auric acid at the same final concentration as the small amount of synthesis, and then reacted at 25 °C for 48 h while shaking. After 48 h, we confirmed the color of each solution with naked-eye observation, quantified them with the same method as outlined above, and evaluated the ability to synthesize a large amount of Au nanoparticles. As shown in Appendix A, the degree of color change was quantified to 1 a.u., 0.5 a.u., and 0 a.u., respectively. We repeated the large amount synthesis 3 times, and added the results of the digitization into a table. The evaluation was based on the total value. The stain with the most comprehensive results for both small amount and large amount synthesis was selected and used as the microorganism to be studied for the rest of this research.

### 2.3. Identification of the Selected Strain

In order to identify the selected strain, we extracted DNA by using the alkaline heat extraction method. Firstly, we took a small amount of colonies on the plate with sterile micropipette tip and suspended them in 50 μL of sterile water. We then added 100 mM NaOH to the 50 μL suspension and mixed them together. Then, we carried out a heat treatment above 95 °C for 10 min. After the heat treatment, we added 11 μL of 1 M Tris-HCl (pH 7.0) and mixed. Finally, the mixture was centrifuged at 12,000 to 15,000 rpm for 1 min, and the supernatant was collected and used as the DNA extraction solution. After the PCR reaction, we performed the electrophoresis for 30 min on a 2% agarose gel to confirm DNA amplification. After confirmation, the solution after the PCR reaction was subjected to ethanol precipitation. Then, we mixed the purified DNA sample solution with the primers 10F, 518F, 800F, and 1500R, respectively, and sent the solution to FASMAC for sequencing.

We then placed the 16S rRNA sequence of the selected strain into the analysis tool “Classifier” provided in the Web tool called RDP (Ribosomal Database Project). The resulting classifications were expressed as “reliability”. If the “reliability”was above 80, then the genus of the strain could be determined. Then, we used the analysis tool called “Sequence Match” to express the related species. After that, we selected the appropriate species, entered the Pairwise Sequence Alignment, input the sequence of the selected strain and the sequence of its related species, and checked the homology score.

The phylogenetic tree capability, which can express the relationship between the selected strain and the selected related species, was created by the software called MEGA (Molecular Evolutionary Genetics Analysis). In addition, a phylogenetic tree assessment was performed and analyzed by using the neighbor-joining method.

### 2.4. Microscopic Observation and Other Analysis

We pipetted the suspension of K-142 cell and dropped it on a glass plate, as the sample to be observed the cell morphology under an optical microscope. Furthermore, we also used TEM (transmission electron microscope) and SEM (scanning electron microscope) to observe the cells of K-142 and the particles which they synthesized. We dropped 2 µL of solution on the grid coated with carbon powder, and used it as a sample for TEM (JEM-1010, JEOL, Tokyo, Japan) observation after drying. TEM observation was performed at an accelerating voltage of 200 kV. In addition, TEM images and electron diffraction of crystals were observed using TEM (JEM-2010, JEOL, Tokyo, Japan). The elemental composition of the observed nanoparticles was analyzed by detecting the characteristic X-rays specific to each element with an energy dispersive X-ray analyzer (EDS) (EMAX, HORIBA, Kyoto, Japan). For SEM observation, the sample was prepared by attaching a carbon tape to an aluminum sample table, adding 2 µL of the solution, and drying. After performing a Pt coating treatment by using ion sputtering (E-1030, HITACHI, Tokyo, Japan), we used ImageJ [13] to measure the particle diameter of the particles observed through electron microscopes and created a histogram. Particles larger than 100 nm were excluded when creating the histogram.

Powder X-ray diffraction (XRD) was performed to examine whether the Au nanoparticles synthesized by K-142 cells had a crystal structure. A total of 48 h after the addition of auric acid to the bacterial cell suspension, we collected the bacterial cells and freeze-dried them overnight. We then crushed them with a pestle/mortar and placed them on a non-reflective plate made of single crystal silicon for the XRD measurement [14]. Measurements were then performed with an X-ray diffractometer (RINT Ultima+, Rigaku, Tokyo, Japan). The results of the measurements are shown in Appendix A.

The ultraviolet-visible spectra of the absorbance at 400 to 700 nm were measured with a spectrophotometer (V-550 spectro photometer, JASCO, Tokyo, Japan). The measurement was performed once for every 1 nm wavelength.

### 2.5. Synthesis of Nanoparticles Using Metal Salt Solution Other Than Au

We weighed AlCl_3_·6H_2_O, CuCl_2_, (CH_3_COO)_2_Pb·3H_2_O, Pb(NO_3_)_2_, AgNO_3_, CdCl_2_, CdSO_4_·8H_2_O, and PtCl_2_ with a balance (AB135-S, METTLER TOLEDO, Columbus, OH, USA) and then dissolved them in the corresponding solvent to prepare the metal salt solution with a final concentration of 50 mM. PtCl_2_ was dissolved in hydrochloric acid and the others were dissolved in ultrapure water. All metal salt solutions were sterilized by autoclaving at 120 °C for 20 min. In addition, the prepared solution PtCl_2_ was orange and transparent, and the other metal solutions were colorless and transparent.

The reaction solution (after 72 h of reaction) of K-142 bacteria and each metal were observed with TEM. In order to analyze the elemental composition of the particles observed with TEM, the characteristic X-rays inherent to each element were detected by using EDS to confirm the element distribution around the cell of K-142. In addition, we used TEM to observe the crystallization and the electron diffraction of crystals in the TEM images.

### 2.6. Measurement of Metal Concentration Efficiency

Each metal solution was added to the K-142 cell suspension fora final concentration of 0.5 mM, and reactions to shaking were tested for 24 h. After the reactions, we centrifuged (4000× *g*, 4 °C, 10 min) the reaction solution and collected the supernatant (the bacterial cells were separated). We filtered the collected supernatant with the 0.45 μm filter, and marked the filtrate obtained here as “with K-142”. We then replaced the cell suspension in the above operation with ultrapure water and then added the metal salt solution in the same manner. The reaction was also performed while conducting shaking, and was followed by filtration. We marked the solution obtained here as “without K-142”.

We decomposed the organic matter in the sample solution and transferred the samples of “without K-142” and “with K-142” to glass test tubes (IWAKI, Shizuoka, Japan) (1 mL each). We then decomposed them with nitric acid. After adding 1 mL of concentrated nitric acid to each test tube, two rounds of wet ashing decomposition were carried out at 120 °C, and each round lasted for10 h. Then, we added 1 mL of hydrogen peroxide to each test tube and treated them at 120 °C for 10 h. The decomposed products were made up to 10 mL with 0.1 M dilute nitric acid and were used as the samples for ICP-AES (inductively coupled plasma atomic emission spectroscopy) measurement. Since the measurement required a metal standard solution, we prepared the 0.1 M nitric acid aqueous solution in advance. Then we added 100 μL for each of the standard solutions of Ag, Cd, Pb, Cu, and Al into the same 100 mL volumetric flask, and diluted the volume to 100 mL with the 0.1 M nitric acid aqueous solution to make a solution with nitric acid as the solvent metal standard solution (1 ppm).

We used ICP-AES (SPS3500, SII NanoTechnology, Chiba, Japan) to measure the metal concentrations under the conditions of Ag (with a wavelength of 338.289 nm), Al (with a wavelength of 396.152 nm), Cd (with a wavelength of 214.506 nm), Cu (with a wavelength of 224.770 nm), and Pb (with a wavelength of 220.353 nm).

### 2.7. Confirmation of Survival State of Bacterial Cells by Fluorescence Observation

After being treated with 0.5 mM of each metal (Ag/Al/Cd/Cu/Pb) salt solution for 24 h, the precipitate was recovered by using centrifugal separation (4000× *g*, 4 °C, 10 min) “with K-142”, and then the precipitate was suspended in 1 mL of sterilized water. Then, we used the suspension as the object of subsequent fluorescence microscope observation.

We mixed SYTO^TM^ 9 and propidium iodideand added the staining reagent FilmTracer^TM^ LIVE/DEAD Biofilm Viability Kit (Thermo Fisher Scientific, Waltham, MA, USA) [15,16] with propidium iodide (the mixing volume ratio was 1:1). We then mixed 3 μL of the mixed staining reagent with 1 μL of the suspension, and let it stand still for 15 min. We dropped 2 μL of the reacted mixture onto the slide glass to prepare the samples. Observation was performed with a confocal laser microscope (FV1200 IX83, Olympus, Tokyo, Japan). We took a differential interference image, green fluorescence image (excitation/emission wavelength 482/520 nm), and red fluorescence image (excitation/emission wavelength 490/604 nm) for each sample.

## 3. Results and Discussion

### 3.1. Screening, Identification, Observation and Analysis of Microbially Synthesized Nanoparticles

All isolated strains served as a collection of microbial strains. Finally, a library of 282 strains (K-1 to K-282) was created. We added auric acid to the culture solution of the isolated microorganisms and observed the color change. In addition, an evaluation based on the presence or absence of red to purple coloration due to the presence of Au nanoparticles was constructed. The evaluation was performed by using a method of synthesizing Au nanoparticles with a small amount and a large amount of bacterial suspension multiple times and digitizing the results. Through this evaluation, the strain with the best synthesis ability was selected.

#### 3.1.1. Strain Screening

The first round of screening targeted all strains in the collection. The strains were transplanted into 24-well plates. The auric acid solution was added to each well. After incubation for two days, the color changes of the solution were observed. The wells that turned purple and the wells with a large degree of color change were selected in this round. The results are shown in the photograph of 24-well plate (Appendix A). Each of the selected strains was marked with a red frame. In this round, we selected 24 strains from 282 strains. Names and the sampling locations of the selected strains are shown in Appendix A. Since the strains selected in this round had clearly changed color after adding auric acid, it was judged that they were more resistant to Au ions and stronger in synthesizing Au nanoparticles than the unselected strains.

The media components can interfere with the reaction and the reproducibility of the synthesis of Au nanoparticles at the large scales of culture. Therefore, we did a second round of screening on the strains which were selected in the first round. The second round was divided into two stages: small amount synthesis (1 mL) and large amount synthesis (50 mL). The evaluation methods and standards were the same as in the first round and based on the results of each reproduction experiment. The digitization examples of small amount synthesis and large amount synthesis are shown in Appendix A. The wells which turned dark purple were recorded as 1, light purple or other colors were recorded as 0.5 a.u., and the wells with no color change were recorded as 0 a.u.

Appendix A shows the digitizational results of small amount synthesis of the target strains (the 24 strains selected from the first round) in the second round. Ten strains (K-19, K-32, K-67, K-87, K-149, K-166, K-167, K-237, K-255, and K-268) showed results of 0 a.u. It was considered that these 10strains produced color changes in the first round because of the promotion of synthesis by medium components. So, we judged that they were “unable to synthesize Au nanoparticles”. Four strains (K-29, K-90, K-140, and K-165) with a total of 0.5 a.u. to 4.0 a.u. were evaluated as having poor reproducibility for the small amount of synthesis. The remaining 10 strains with results of 4.5 a.u. or more (K-13, K-49, K-62, K-86, K-98, K-108, K-119, K-123, K-141, and K-142) were evaluated as having good reproducibility for the small amount of synthesis and we continued to do a large number of synthesis experiments with these 10 strains. Appendix A shows the digitizational results of a large amount of synthesis. The total value of the digitizational results of small and large amounts of synthesis as the comprehensive evaluation of each strain was recorded (Figure 1). K-142 was the strain with the highest total value, with 7 times the amount of small amount synthesis and 3 times of the amount of large amount synthesis compared to the other strains. Therefore, it was considered that the strain K-142 has a good reproducibility when synthesizing Au nanoparticles, and has the highest synthetic ability among the strains. K-142 was considered to be the excellent strain that we want, and was used as the target strain in the following experiment.

It is known that the LB-Glucose medium used in this screening has a simple medium composition and is a versatile and inexpensive medium. We decided to use this LB-Glucose medium, as it then becomes possible to select bacteria which can be cultivated simply and inexpensively, and obtain bacteria that can be used in practical industries and bioremediation applications. In the process of screening strains that are resistant to auric acid and synthesize Au nanoparticles, many factors could affect the growth of bacteria, which may alter the outcome of Au nanoparticle synthesis. For example, conditions such as temperature, aerobic or anaerobic, inoculum size, and culture time could all have impacts. Since it is not possible to consider all of the differences in conditions depending on all bacterial species, we chose a temperature of 25 °C, which is generally suitable for the growth of microorganisms, as well as the oxygen concentration which occurs under normal aerobic conditions. In addition, we made the inoculation amount as uniform as possible, and set the cultivation time as a range from 1 day to 7 days that is generally required for growth of microorganisms.

Previous reports have suggested that in the presence of large amounts of amino acids and sugars, Au nanoparticles can be produced from auric acid, even in the absence of microorganisms [17,18,19]. In previous reports on the synthesis of Au nanoparticles using microorganisms, auric acid was often added when the medium and cells were mixed [20]. However, it is difficult to evaluate and judge under the interference of medium components. In our study, we designed the screening portion to occur after removing the medium components. Since the metabolites produced by bacteria change under various conditions, the reproducibility of Au nanoparticle synthesis was emphasized in the screening stage. We solved the above problem by digitizing the degree of the color change and comparing the values from multiple experiments. After verification of reproducibility, a strain that can stably synthesize Au nanoparticles were finally selected through comprehensive evaluation.

#### 3.1.2. Identification of Selected Strain

The 16S rRNA sequence of the screened strain K-142 of was obtained by PCR using universal primers and DNA sequencing. As the result of the identification of the genus, the score was 100%, indicating that K-142 belongs to the *Bacillus* genus. We obtained 10 species with high homology from the strains registered as type species in NCBI using RDP, and created a phylogenetic tree by the neighbor-joining method using MEGA software. The results of phylogenetic tree analysis are shown in Figure 2. It showed that *Bacillus*
*butanolivorans* was the closest species to K-142. As the result of observing the shape of the cells with an optical microscope, we found that K-142 appeared as bacilli (Appendix A). In the long-term culture, we found that the colony of K-142 is very viscous and that K-142 is not easy to separate from liquids such as water and liquid culture. In the daily culture process, we found that the colony of K-142 is larger and thicker than other strains in the same cultivation time, and the colony of K-142 can be stored at 4 degrees for about half a year but would still synthesize Au nanoparticles normally (data not shown).

*Bacillus* bacteria are aerobic (partially facultative anaerobic) gram-positive bacilli that form spores, and some strains belonging to *Bacillus* have the ability to convert atmospheric nitrogen into other nitrogen forms, such as ammonia. This genus includes *Bacillus*
*subtilis*, which is important as a model organism for gram-positive bacilli. The *Bacillus* is a genus that is ubiquitous in water and soil and contains a large number of species. Among them, many species are adapted to various extreme environments that possess high pH, low temperatures, high salt concentrations, and high pressure. A subspecies of *B*. *subtilis* mentioned previously is *Bacillus*
*natto*. Especially in recent years, synthesis of Ag nanoparticles has included using extracellular polysaccharide of *B*. *subtilis* [21], synthesis of Ag nanoparticles by *Bacillus* sp. [22], or synthesis of Pb nanoparticles by *Bacillus*
*toyonensis* SCE_1_ [23]; there have been many reports on the study of synthesis of nanoparticles with bacteria of the *Bacillus* genus.

#### 3.1.3. Observation and Analysis of Nanoparticles Synthesized by K-142 Cells

In the screening stage, we used a single concentration of auric acid for experiments. We changed the concentration of auric acid to explore its synthesis ability. The auric acid solution was added to the suspension of K-142 bacterial cells so that the final concentrations were 0 mM, 0.2 mM, 0.4 mM, 0.6 mM, 0.8 mM, and 1.0 mM. We also assessed auric acid without the presence of K-142. After the auric acid addition and the reaction with shaking for 48 h, we confirmed the color change of the suspension. The results are shown in Figure 3a. In the group of “with K-142”, the purple color of the solution became darker gradually depending on the concentrations of auric acid varying from 0.2 to 1.0 mM. The control group, which was marked as “without K-142”, showed that a color change would not occur when there was only auric acid, and the solution would therefore still present the original light yellow of auric acid. The above results suggested that any purple change came from both the presence of the bacteria K-142 and auric acid.

UV-Vis spectra were measured in the wavelength range of 400 to 700 nm for a solution with different concentrations of auric acid, which was marked as “with K-142” in Figure 3a (Figure 3b). The absorption maximum wavelength of the solution that turned purple was about 550 nm, which was in agreement with the characteristic wavelength of the SPR of Au nanoparticles.

In order to confirm whether the color change after the addition of auric acid was due to Au nanoparticles, we observed the samples using SEM. Appendix A shows the SEM image of K-142. Appendix A is an image without the addition of auric acid where rod-shaped microorganisms can be observed, similarly to the results observed with the optical microscopes. Appendix A is the results with auric acid (auric acid concentration 1.0 mM); we found that white microorganisms were formed around the cells. Appendix A is an enlarged view of the results with auric acid, and spherical particles; a small number of triangular particles were observed. The spherical particles had a size of about 25 nm, while the triangular particles were larger than the average size of the spherical particles.

Figure 4a,b show the TEM images of the K-142 cells and the synthesized nanoparticles. Circular and triangular black dots were observed around the gray cells. Based on the analysis of the elemental composition of the black particles in Figure 4a,b with the attached EDS device (Figure 4d), the synthesized nanoparticles consisted of Au elements. Figure 4e shows the electron diffraction pattern of nanoparticles synthesized by K-142 cells. It also shows the ring pattern of Au crystals. We calculated the area of the nanoparticles seen in the picture in the upper right corner of Figure 4c, and created a histogram of their diameters. From the results shown in Figure 4c, the particle sizes were distributed in the range of 10–60 nm, and the average particle size was 25.9 ± 5.5 nm.

We freeze-dried the bacterial cells (synthesized with Au nanoparticles) overnight, crushed them with a pestle/mortar, and placed them on a non-reflective plate made of single crystal silicon for the measurement. The XRD pattern of the obtained K-142 powder and the spectrum of the Au crystal obtained from AMCSD (the American Mineralogist Crystal Structure Database) [24] are respectively shown in Figure 5. Peaks were detected at 2θ = 38.2°, 44.4°, 64.6°, 77.6°, and 81.8°. These peaks coincided with the peaks in the crystalline surface at {111}, {200}, {220}, {311}, and {222} of Au registered in the database. This means that the nanoparticles synthesized by K-142 are Au crystals.

### 3.2. Synthesis of Nanoparticles with Metal Salt Solution Other Than Au

When K-142 is actually used in bioremediation (such as for the recycling of heavy metals in the environment), it is not enough to limit the reduction of only one metal (such as Au) to form nanoparticles. Therefore, we investigated the response of K-142 to metals other than Au. Metal elements are indispensable to living organisms, but if they exceed a certain value due to ingestion or accumulation, then they are considered to be harmful to living organisms. Both AgNO_3_ and Ag nanoparticles are highly toxic substances for bacteria [25]. Previous study shows that the viability of cells exposed to suspensions of Ag nanoparticles containing high concentrations of Ag ions is halved [26]. In addition, the results of animal experiments in which Ag nanoparticles induced destruction of the blood-brain barrier and swelling of astrocytes in the brain and caused neurodegeneration have been reported [27]. It is also known that Pb, Cu, and Cd may enter the living body excessively through the skin or by inhalation to cause acute or chronic poisoning. High levels of Pb in the blood may impair the reproductive function of animals [28]. Cu in drinking water may greatly increase the incidence of animal diseases (such as Alzheimer’s disease) [29]. Inhalation of air contaminated with Cd can cause severe respiratory effects and induce pneumonia [30]. Ingestion of foods contaminated with Cd can cause acute gastrointestinal disorders such as vomiting and diarrhea [31]. Long-term exposure to Cd can cause kidney damage [32]. Many studies and reports have indicated that excessive intake of Cd can have toxic effects on many organs. Although Al has long been considered to be non-toxic, it has been reported that an overdose of Al can induce neurodegeneration [33]. A report was also published stating that the onset of Alzheimer’s disease is potentially associated with the long-term intake of aluminum hydroxide [34], while the actual cytotoxicity of aluminum remains controversial.

For this section, we used the metal salt solution mentioned above to act on K-142 and discussed the results of the reaction. The metal salt solution of Ag, Al, Cu, Pt, Cd, and Pb was added to the bacterial cell suspension of K-142, and shaken for 72 h to try to synthesize the aforementioned metal nanoparticles. In experiments using auric acid, the approximate concentration range of the reaction was already known, but it was not clear which metal corresponds to the appropriate concentration. Therefore, for the metals other than Au, it is better to use a wider concentration range than when using auric acid (0.2 to 1.0 mM). Therefore, a metal salt solution was prepared in the range of 0.1 mM (low concentration) to 5.0 mM (high concentration). In order to confirm whether nanoparticles were synthesized, we conducted a visual observation of color change and a TEM observation.

#### 3.2.1. Visual Observation of Changes

The photos of the sample solution after addition immediately and 72 h later are shown in Appendix A. The sample added with the colorless AgNO_3_ aqueous solution turned orange after 72 h (Appendix A). It is known that during the synthesis of Ag nanoparticles, the color of the solution changes to yellow or orange due to SPR [12], so it has been speculated that Ag nanoparticles were synthesized due to K-142. Similarly, 72 h after the addition of a colorless AlCl_3_ aqueous solution, there was no change in color, but bacterial cells were precipitated (Appendix A). In the blue CuCl_2_ aqueous solution and the dark orange PtCl_2_ hydrochloric acid solution, the color of the metal solution appeared to be dark in proportion to the concentration of the metal salt, but there was no significant difference between the color immediately after the addition and 72 h after the addition (Appendix A). The sample solution containing colorless CdCl_2_ and CdSO_4_as well as colorless (CH_3_COO)_2_Pb and Pb(NO_3_)_2_ aqueous metal solutions remained colorless and experienced no color changes (Appendix A). These results were different when synthesizing Au and Ag nanoparticles, where the color change of the solution that occurs can be seen to be due to SPR. For Cd, Cu, Al, and Pt, it is difficult to judge them only by the color of the solution before and after the reaction. Therefore, we observed the suspension after the reaction caused by TEM to confirm whether metal nanoparticles were synthesized or not.

#### 3.2.2. Observation by TEM

As shown in Appendix A, only the bacterial cells were observed in the samples added with AlCl_3_, PtCl_2_, and CuCl_2_ solutions, but no particles were observed. In the sample added with AlCl_3_, we found that the two ends of the bacterial cells appeared to be darker than the middle part (Appendix A). When the hydrochloric acid solution of PtCl_2_ and the CuCl_2_ aqueous solution were added, there was no change in the appearance of the bacterial cells (Appendix A).

The results of adding aqueous solutions of AgNO_3_, CdSO_4_, and Pb(NO_3_)_2_ are shown in Figure 6. Large particles with various shapes were observed at low concentrations (0.5 mM) of AgNO_3_ (Figure 6a). With the addition of a high concentration (2 mM to 5 mM) of AgNO_3_ solution, particles with smaller diameters were observed in the visual field (Figure 6b–d). These results showed that K-142 can synthesize nanoparticles in a wide range of Ag ion concentrations. With the addition of CdSO_4_ aqueous solution (0.2 mM), nanoparticles were observed around the cells (Figure 6e). At 1.0 mM, it was observed that nanoparticles were not synthesized and some bacterial cells were destroyed (Figure 6f). Nano-sized particles were observed under the conditions of Pb(NO_3_)_2_ concentrations of 0.1 mM, 0.2 mM, 0.5 mM, and 1.0 mM (Figure 6g–j). At the low concentrations (0.1 mM and 0.2 mM), deposition appeared on a few peculiar bacterial cells (Figure 6g–h). A large number of needle-like crystals were formed in the entire range of these bacterial cells. At the concentrations of 0.5 mM and 1.0 mM, all of the synthesized nanoparticles aggregated inside the cells, and no particles were found outside the cells (Figure 6i–j). At 0.5 mM, both ends of the cells were seen to be darker than the middle parts (Figure 6i).

To investigate the elemental analyses, we used EDS analyses in TEM imaging. As a result of elemental analysis using EDS, Cu peaks derived from grids were observed in all of the samples (Figure 7). In addition to the Cu peak, peaks of the elements Ag, Si, P, and Cl were detected in the sample to which AgNO_3_ (3.0 mM) was added (Figure 7a).In the sample to which CdSO_4_ was added (0.2 mM), peaks of the elements Cd, O, Si, P and S were detected (Figure 7b).In the sample to which Pb(NO_3_)_2_ (0.2 mM) was added, the peaks of the elements Pb, O, Si, S, and K were detected, and the peak of the element S and the peak of Pb near 2 keV were found to overlap (Figure 7c). In the sample to which Pb(NO_3_)_2_ (0.5 mM) was added, the peaks of the elements Pb, O, Si, and S were detected, and it was seen that the peak of the element S and the peak of Pb overlap with each other at around 2 keV (Figure 7d).

In the TEM image of each sample, the electron diffraction pattern of the particle indicated by the arrow was compared with the ring pattern in the database to identify the mineral. In the sample which AgNO_3_ (3.0 mM) was added to, the ring patterns coincided with the crystal planes {111}, {200}, {220}, and {311} of Ag (Figure 7a). In the sample which CdSO_4_ (0.2 mM) was added, the ring pattern coincided with the crystal planes of CdS {111}, {200}, {220}, and {311} (Figure 7b). In the samples which Pb(NO_3_)_2_ (0.2 mM and 0.5 mM) were added to, the ring patterns were consistent with the PbS crystal planes {111}, {220}, {311}, and {024} (Figure 7c,d). Combined with the results of EDX analysis elements, the synthesized nanoparticles were Ag crystals when AgNO_3_ was added to the concentration of 3.0 mM, while the synthesized nanoparticles were CdS crystals when CdSO_4_ was added to a concentration of 0.2 mM. When Pb(NO_3_)_2_ was added to the concentration of 0.2 mM and 0.5 mM, the synthesized nanoparticles were PbS crystals.

K-142 can reduce Ag ions in a wide range of concentrations to Ag. Under lower concentration conditions (less than 0.5 mM), large crystals are synthesized, and under higher concentration conditions, particles with smaller particle diameters are synthesized. The opposite occurs for Au (low concentration Au ions are reduced to nanoparticles, while high concentration Au ions are reduced to bulk Au). Ag has been used as an antibacterial material in many previous studies [35,36,37]. In other words, some microorganisms cannot survive in the presence of Ag ions, let alone maintain the ability to synthesize Ag nanoparticles. Therefore, the result that K-142 can synthesize nanoparticles in high concentrations of Ag ions is also considered to be one of the evidences that K-142 is strong to heavy metals. The nanoparticles synthesized by K-142 with AgNO_3_ were not Ag_2_O or Ag_2_O_2_ but instead were an Ag simple substance. Although Ag is quickly oxidized into Ag_2_O or Ag_2_O_2_ under aerobic conditions, it can become particles that are difficult to oxidize due to the actions of the K-142.

Although cells were destroyed when Cd ions were added to the culture solution at a higher concentration (1.0 mM), nanoparticles during this experiment were successfully synthesized under the condition of low concentration (0.2 mM). This synthetic trend was consistent with the results of Au nanoparticles. The synthesized nanoparticles were predicted to be CdS based on the elemental analysis and electron diffraction patterns. In living organisms, soft metals (such as Cd) are often used in conjunction with sulfur compounds for detoxification. Therefore, it is considered that CdS exists inside and outside the bacterial cells. With the addition of Pb ions, needle-like nanoparticles were synthesized intracellularly. These needle-like nanoparticles were very uneven in size, varying from several nm to 500 nm. From the results of elemental analysis and electron diffraction, it was considered that they were PbS crystals. It is considered that PbS minerals are formed in the living body because the sulfur compounds that are bound to Pb are the same as the ones that are bound to Cd. In the elemental analysis results of all Ag, Cd, and Pb samples, it is considered that phosphorus comes from microbial organic matter, while silicon may come from dust in the air.

### 3.3. Measurement of Metal Concentration Efficiency

From the electron microscope observations in Section 3.2.2, changes in the contrast and thickness of K-142 cells were observed after the addition of metal ions. It was observed that the ends of the cells became black (Figure 7c,d).This phenomenon was considered to be the result of K-142 accumulating metal ions in the cells. Therefore, we conducted experiments on the ability of K-142 to accumulate the metals Ag, Al, Cd, Cu, and Pb. The AgNO_3_, AlCl_3_, CdSO_4_, CuCl_2_, and Pb(NO_3_)_2_ solution was added to the K-142 bacterial cells suspension to a final concentration of 0.5 mM. The supernatant was decomposed with concentrated nitric acid, the decomposition product was dissolved in dilute nitric acid, and the metal concentration was measured by using ICP-AES. The above-mentioned samples were marked as “with K-142”. Notably, we used water instead of bacterial cell suspension to make the samples marked as “without K-142”, and also measured the metal concentration by using ICP-AES.

The results measured by using ICP-AES are shown in Figure 8. Comparing the metal concentrations of the samples “without K-142” and “with K-142”, we evaluated the abilities of K-142 to accumulate metals according to the concentration of K-142 bacteria compared to the salt solution made up of Ag, Al, Cd, Cu, and Pb. The concentration of Ag ions dropped by 99.2% due to the presence of K-142 (Figure 8a(i)). In addition, the concentration of Al ions decreased by 98.2% (Figure 8b(i)), the concentration of Cd ions decreased by 83.9% (Figure 8c(i)), the concentration of Cu ions decreased by 64.8% (Figure 8d(i)), and the concentration of Pb ions decreased by 97.4% (Figure 8e(i)). Based on these results, it was considered that K-142 greatly contributed to metal accumulation.

The results showing that the concentration of each metal has dropped by more than 60% prove that K-142 is likely to deposit metals. The concentration efficiency of Cu was the lowest (64.8%), and it was considered that accumulation is difficult by the living body because Cu is a borderline metal, because borderline metals are not precipitated by oxidation and sulfidation reactions.

### 3.4. Confirmation of Cell Viability by Fluorescence Observation

In order to investigate whether K-142 is still alive after treatment with each metal ion (Ag^+^, Cd^2+^, Pb^2+^, Cu^2+^, and Al^3+^), the life and death of the bacterial cells were observed after staining with fluorescent reagents. The differential interference images, green fluorescence images (emission wavelength 520 nm), and red fluorescence images (emission wavelength 604 nm) of each sample are shown in Figure 8.

When AlCl_3_ was added to the K-142 culture solution, both green and red fluorescence were observed, indicating that the bacterial cells were dead (Figure 8b(iv)). With the addition of CdSO_4_, signals were observed in green fluorescence and red fluorescence but the signals in red fluorescence were very weak, suggesting that they may be alive but their vitality has decreased (Figure 8c(iv)). In the presence of AgNO_3_, CuCl_2_ and Pb(NO_3_)_2_, only green fluorescence was observed. On the other hand, there was no red fluorescence signals at all (Figure 8a(iv),d(iv),e(iv)). From these results, the bacteria K-142 could be seen to be completely alive under the condition of Ag, Cu, and Pb ions.

In order to confirm the viability of K-142 after treatment with the metal solution, we treated the cells with the FilmTracer^TM^ LIVE/DEAD Biofilm Viability Kit and observed it under a fluorescence microscope. The FilmTracer^TM^ LIVE/DEAD Biofilm Viability Kit is capable of performing assays with two-color fluorescence corresponding to bacterial viability based on membrane permeability. SYTO^TM^ 9is a membrane-permeable green fluorescent nucleic acid stains, and propidium iodide is a membrane-impermeable red fluorescent nucleic acid stain. When SYTO^TM^ 9 is used alone for staining, it will generally mark all the bacteria in the population at the same time. It will mark both bacteria with undamaged cell membranes and bacteria with damaged cell membranes. In contrast, propidium iodide only penetrates into bacteria through damaged cell membranes. After treating K-142 with Ag, Cu, and Pb ions, the cell membrane was not damaged. After treatment with Cd ions, the damage to the cell membrane was small. This indicates that K-142 has strong resistance to Ag, Cu, Pb, and Cd ions. The result about the addition of Al ions showed that the cell membrane of K-142 was heavily damaged and the bacteria died after the treatment.

## 4. Conclusions

In this study, we successfully screened microorganisms with a strong ability to synthesize Au nanoparticles from environmental water. Through the identification of selected strain from more than 282 strains, it was confirmed that the selected K-142 is the bacterium belonging to the *Bacillus* genus. K-142 induced the synthesis of Au nanoparticles that have a 25.9 nm average size. K-142 can not only synthesize Au nanoparticles, but can also synthesize Ag, CdS, and PbS nanoparticles, and showed the excellent deposition efficiency of the several metal ions such as Ag^+^, Al^3+^, Cd^2+^, Cu^2+^, and Pb^2+^ (Figure 9). In addition, the results of fluorescent staining showed that K-142 could survive after being treated with a metal solution for 1 day. Therefore, K-142 is expected to be useful for bioremediation due to its ability to synthesize Au and other metal nanoparticles and its excellent ability to accumulate metals.

## Figures and Tables

**Figure 1 materials-13-04922-f001:**
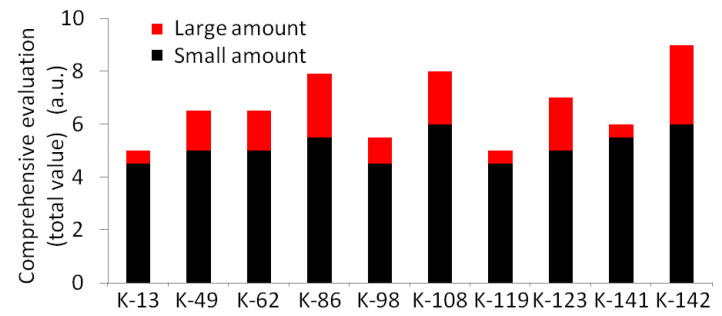
Total value of small amount synthesis and large amount synthesis. The higher the total value, the better the reproducibility of microorganisms.

**Figure 2 materials-13-04922-f002:**
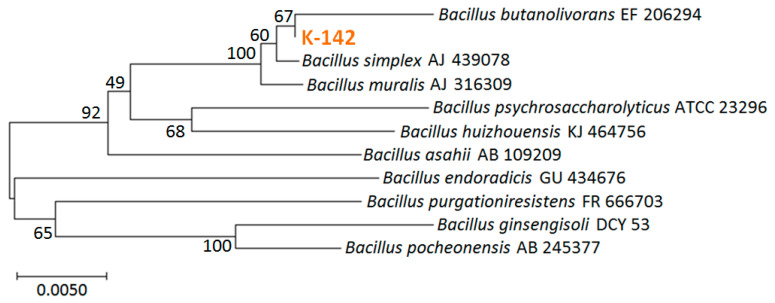
The phylogenetic tree analysis of K-142.

**Figure 3 materials-13-04922-f003:**
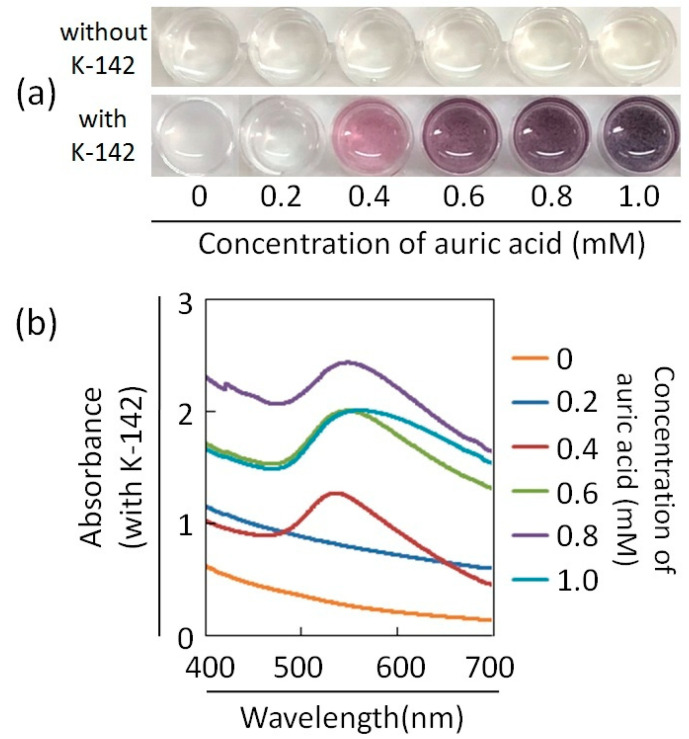
(**a**) Color change of bacterial cell suspensions (small amount) after addition of different concentrations of auric acid. (**b**) UV-Vis spectra of suspensions in Figure 4a. The concentration of K-142 was 0.3 g/mL.

**Figure 4 materials-13-04922-f004:**
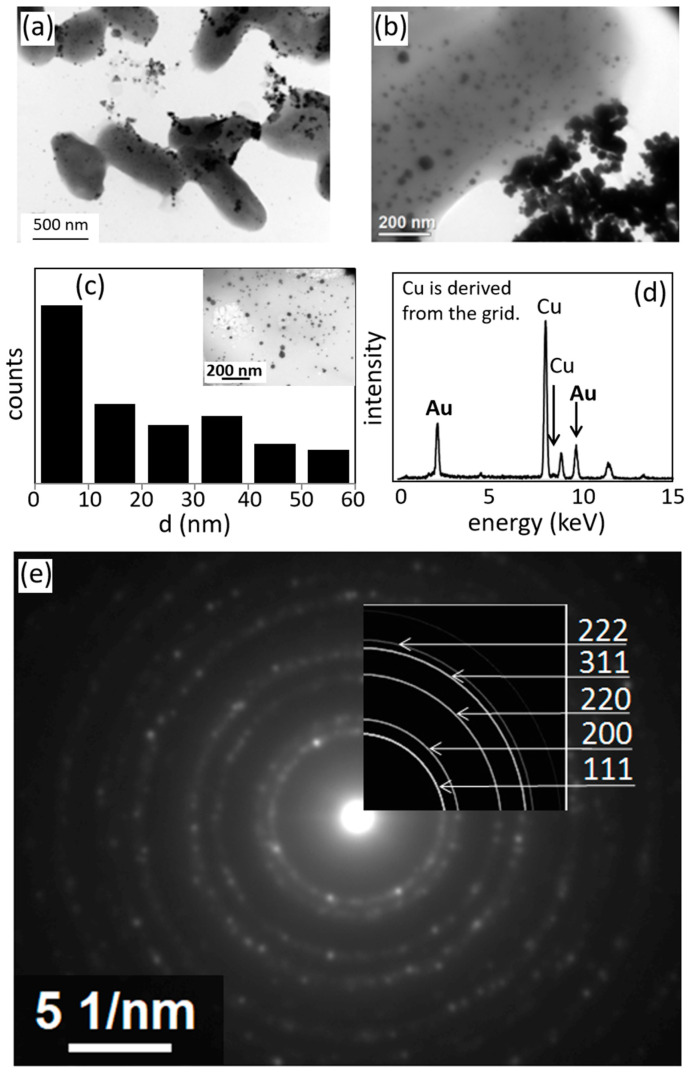
(**a**) Transmission electron microscope (TEM) image of Au nanoparticles synthesized by K-142 and the cells of K-142. (**b**) The enlarged TEM image. (**c**) Histogram of Au nanoparticle size in the view field of the picture in the upper right corner. (**d**) EDS element analysis result of the nanoparticles. (**e**) Electron diffraction pattern of Au nanoparticles.

**Figure 5 materials-13-04922-f005:**
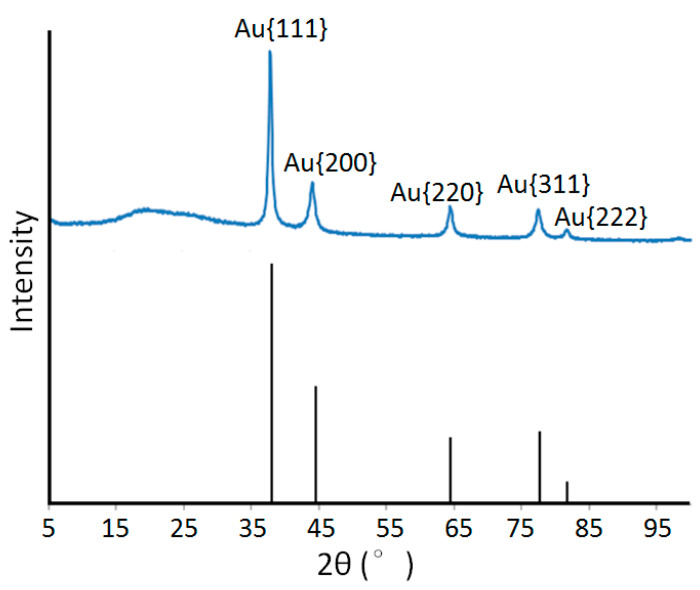
The result of X-ray diffraction spectrum. The blue line: the dry powder of K-142 cell with auric acid. The black line (used as a comparison): The Au crystal standard obtained from the American Mineralogist Crystal Structure Database (AMCSD).

**Figure 6 materials-13-04922-f006:**
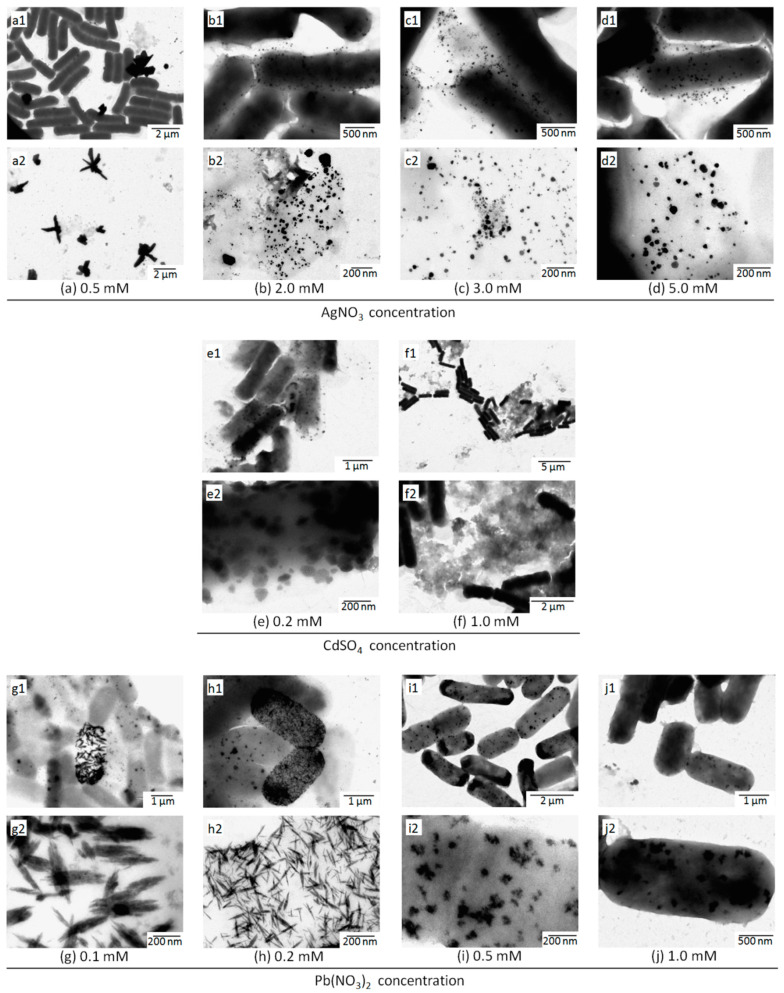
TEM images after the addition of metal salt solution, including (**a**–**d**) AgNO_3_ (aq), (**a**) 0.5 mM, (**b**) 2.0 mM, (**c**) 3.0 mM, and (**d**) 5.0 mM. (**a1**–**d1**) showed the images of bacterial cells and particles synthesized by the bacteria K-142. (**a2**–**d2**) showed the enlarged observation images of the synthesized particles. (**e**–**f**) CdSO_4_ (aq), (**e**) 0.2 mM, (**f**) 1.0 mM. (**e****1**–**f1**) showed the images of bacterial cells and particles synthesized by the bacteria K-142. (**e2**–**f2**) showed the enlarged observation images of the synthesized particles. (**g**–**j**) Pb(NO_3_)_2_ (aq), (**g**) 0.1 mM, (**h**) 0.2 mM, (**i**) 0.5 mM, and (**j**) 1.0 mM. (**g1**–**j1**) showed the images of bacterial cells and particles synthesized by the bacteria K-142. (**g2**–**j2**) showed the enlarged observation images of the synthesized particles.

**Figure 7 materials-13-04922-f007:**
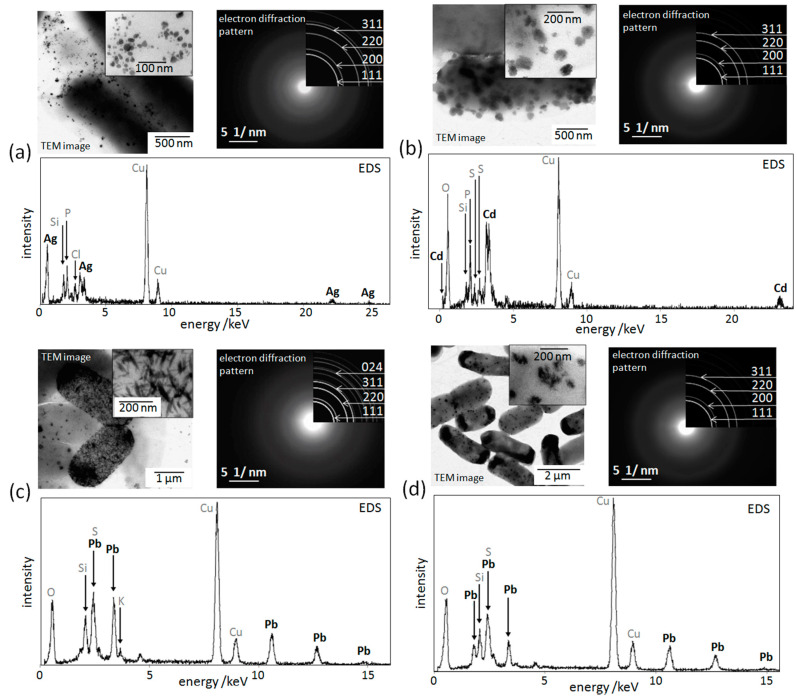
TEM images, electron diffraction patterns of the synthesized particles, and EDS of the synthesized particles of the samples after the addition of a different metal salt solution. (**a**) AgNO_3_ (final concentration 3.0 mM). (**b**) CdSO_4_ (final concentration 0.2 mM). (**c**) Pb(NO_3_)_2_ (final concentration 0.2 mM). (**d**) Pb(NO_3_)_2_ (final concentration 0.5 mM). The results of the electron diffraction patterns and EDS showed that the synthesized nanoparticles were (**a**) Ag crystals, (**b**) CdS crystals, and (**c**,**d**) PbS crystals.

**Figure 8 materials-13-04922-f008:**
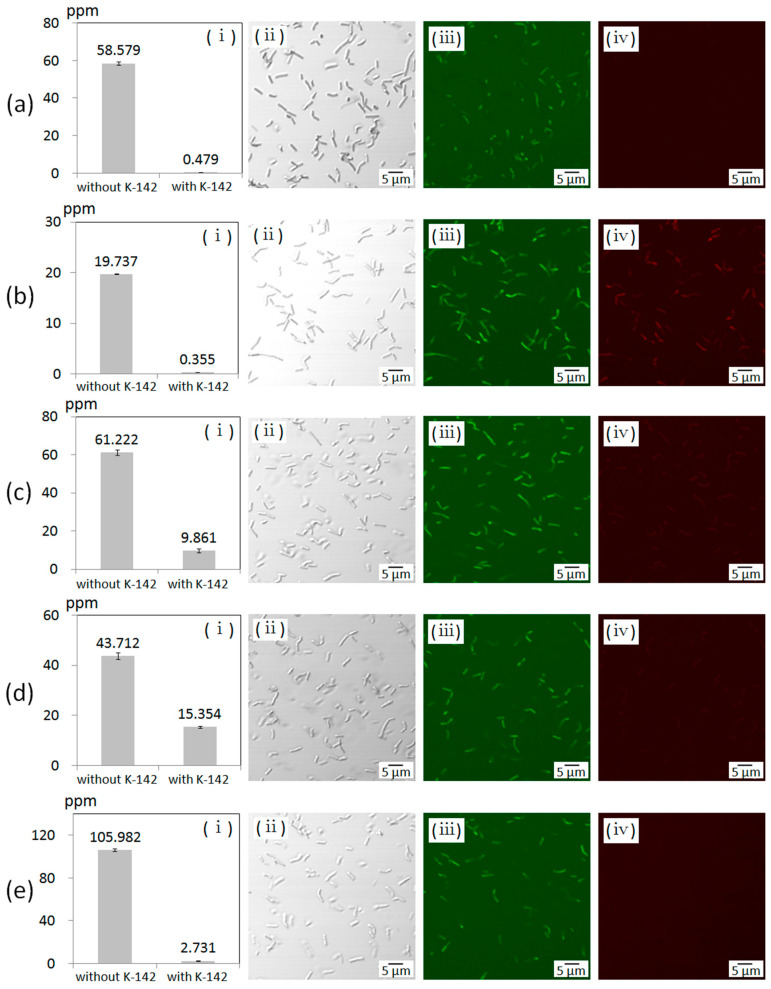
(**a**) K-142 cells treated with 0.5 mM AgNO_3_. (**b**) K-142 cells treated with 0.5 mM AlCl_3_. (**c**) K-142 cells treated with 0.5 mM CdSO_4_. (**d**) K-142 cells treated with 0.5 mM CuCl_2_. (**e**) K-142 cells treated with 0.5 mM Pb(NO_3_)_2_. (**i**) The metal concentrations of the samples (**a**–**d**) “without K-142” and “with K-142” measured by ICP-AES. (n = 3) (**ii**–**iv**) Optical microscope images of K-142 bacterial cells after treatment with metal salt solution and fluorescent staining. (**ii**) Differential interference images. (**iii**) Fluorescent images (Em. 520 nm). (**iv**) Fluorescent images (Em. 604 nm).

**Figure 9 materials-13-04922-f009:**
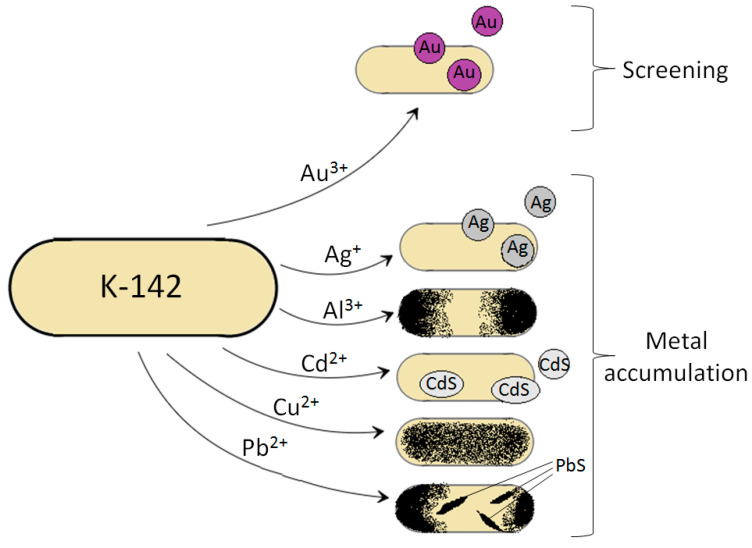
A schematic diagram of the metal accumulation process in K-142.

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
