# Peer review of "Metal Accumulation Using a Bacterium (K-142) Identified from Environmental Microorganisms by the Screening of Au Nanoparticles Synthesis"

_materials, 2020, doi:10.3390/ma13214922_

Round 1
Reviewer 1 Report
This is an interesting topic for biological water purification using Bacillus sp; However, it is still far from publication in my opinion.
1) Generally, there should not be abbreviation (sp.) in the title.
2) Figure 1, 2, 3, 4, 7,11, can be possibly moved to supporting information.
3) The color results are obvious, but please give quantitive values by measuring , for example, the UV-vis spectra. The degree of color change is not a scientific method for quantification. In my opinion, replace all the digitizing color change values with UV-vis results.
4) Figure 5, please change the comprehensive evalution (total value) to the quantitive ICP-MS results.
5) Please combine figure 12,13 and 14.
6) In the legend of Figure 16, please complete with the concentrations used for AgNO3, AlCl3, CdSO4, etc.
7)Figure 8, it is missing the control of auric acid without K-142. In the legend, the concentration of the K-142 should be included.
8) Figure 9, gold NPs were observed both inside the bacteria and outside. There is big aggregation observed, for example in Fig 9b. It is clear that there are some gold NPs more than 100nm. How did you conclude as the particles size was 25.9 nm? How are the nanoparticles formed outside the bacteria?
9) Please correct the grammar mistakes and spell check is required.
Author Response
Reviewer1
Comments and Suggestions for Authors
This is an interesting topic for biological water purification using Bacillus sp; However, it is still far from publication in my opinion.
(response)
Thank you for your meaningful comments and suggestions. We improved and modified our manuscript according to your advice.
- Generally, there should not be abbreviation (sp.) in the title.
(response)
Thank you for your advice. We moved the “sp.”, and changed the title as below.
“Metal accumulation using a bacterium (K-142) identified from environmental microorganisms by the screening of gold nanoparticles synthesis.”
2) Figure 1, 2, 3, 4, 7,11, can be possibly moved to supporting information.
(response)
Thank you for your advice. We have tried to move figure 1, 2, 3, 4, 7 and 11 to supplement data.
3) The color results are obvious, but please give quantitive values by measuring , for example, the UV-vis spectra. The degree of color change is not a scientific method for quantification. In my opinion, replace all the digitizing color change values with UV-vis results.
(response)
Thank you for your meaningful comments. We agree with your opinion. We should give quantitative values by measuring such as the UV-vis spectra. However, to measure the UV-vis spectra of solution, we need to break the cells by ultra-sonication and remove the cell precipitates. Since this process took much time, it is not suitable for the screening process. To identify the active bacterial strain from the library, it is necessary to develop the rapid screening method. From these reasons, we used the digitizing color data in this paper.
4) Figure 5, please change the comprehensive evalution (total value) to the quantitive ICP-MS results.
(response)
Thank you for your meaningful advice. We should know the concentration of Au in the solution at Figure 5. However, to measure the concentration of Au in the solution by ICP-MS, we must do nitric acid dissolution for the supernatant. Since the nitric acid dissolution needs high temperature and much time, it is not suitable for the screening process.
5) Please combine figure 12,13 and 14.
(response)
Thank you for your advice. We have combined figure 12,13 and 14, and modified the corresponding position in the manuscript.
6) In the legend of Figure 16, please complete with the concentrations used for AgNO3, AlCl3, CdSO4, etc.
(response)
Thank you for your correction. Now the description of the concentration has now been added to the figure legend. The concentration of AgNO3, AlCl3 and CdSO4, CuCl2 and Pb(NO3)2 were 0.5 mM in both the ICP measurement and the experiment of treating bacteria with metal ions.
7)Figure 8, it is missing the control of auric acid without K-142. In the legend, the concentration of the K-142 should be included.
(response)
Thank you for your suggestion. We have added the spectrum of the control of auric acid without K-142 to the figure. We also added the concentration of K-142 to the figure legend.
8) Figure 9, gold NPs were observed both inside the bacteria and outside. There is big aggregation observed, for example in Fig 9b. It is clear that there are some gold NPs more than 100nm. How did you conclude as the particles size was 25.9 nm? How are the nanoparticles formed outside the bacteria?
(response)
Thank you for your comment. The average particle size (25.9 nm) is the result of counting and calculating from the field of the picture in the upper right corner of (c). The magnified TEM image (b) showed that the aggregations were composed of nanoparticles. After drying for TEM observation, these gold nanoparticles outside the cells were aggregated as artifact. We guess that the cellular components give the electron to Au3+ and make the gold nanoparticles on the cell membrane.
9) Please correct the grammar mistakes and spell check is required.
(response)
Thank you for your advice. We corrected the grammar mistakes and spell check carefully in the manuscript. After revision process, we will submit our manuscript to English editing service.
Reviewer 2 Report
Comments and Suggestions for Authors:
The paper titled “Metal accumulation using Bacillus sp. identified from environmental microorganisms by the screening of gold nanoparticles synthesis” deals with the investigation and screening of the microorganisms isolated from environmental water by quantifying the reproducibility of synthetic gold nanoparticles and the ability of large-amount synthesis.
The manuscript is well structured in general. The analytical methods which are used at the present study are numerous and significant. Nevertheless, the literature should probably be enriched. Twenty-nine references is a limited number for a study which includes all these interesting scientific methods. Moreover, there are few relatively recent references. It would be auspicable if more recent references were added, or at least if the authors would explain the reason for that.
Another important issue is that in the Introduction part the authors do not present clearly the objectives of the present study. They may describe what was carried out but, unfortunately, they do not make clear what was the need of fulfilling this study as well as which are its novelty aspects. This is important for every scientific paper, so please consider and revise.
The “Results and Discussion” part could be improved with more figures, tables or the presentation of data in diagrams in order the manuscript becomes more interesting to the reader. The “Conclusions” part should definitely be enhanced so that it meets the requirements for publication in such a significant international journal like “Materials”.
As for the English language style, although I do not feel qualified to judge about it probably requires some editing and needs to be improved throughout the manuscript.
For the above reasons major revision should be carried out before publication.
Some comments and suggestions for the authors concern the abstract, at first. The chemical elements are there referred by their full names while at the rest of the manuscript they are listed by their symbols which is more often used. Please consider and revise so that all elements are listed in a uniform manner throughout the whole text.
Figure 1:
The authors could provide more information regarding the study area, in order to make it more clear to the reader where exactly it is located. Apart from the images that are already depicted, probably an image of the study area’s position in Japan would help.
Materials and Methods:
At this part the authors describe the methodologies that they followed for the accomplishment of the present study. Unfortunately they do not mention the researchers by which these methodologies were applied for the first time. Meaning, according to whom researchers are these methodologies applied? No references are mentioned following each methodology. Unless they are applied here for the first time. If this is the reason, please state and clarify that. If not, please add the corresponding references.
Figure 5:
The vertical axe “y” of the diagram presents the comprehensive evaluation (total value). What is the unit of this value? The unit by which a value is evaluated must always be mentioned. Please clarify that. The same should probably be considered for the “amount” in the columns. In what way these parameters are quantified? Please consider an make that clearer to the reader.
Figure 8b:
Please add the corresponding reference by which the presented parameters (wavelength, absorbance) were derived. Respective bibliographic references should probably be added for Figures 9, 10, 15, 16, as well.
Conclusions:
As mentioned above, this part needs definitely to be improved.
Author Response
Reviewer2
Comments and Suggestions for Authors:
The paper titled “Metal accumulation using Bacillus sp. identified from environmental microorganisms by the screening of gold nanoparticles synthesis” deals with the investigation and screening of the microorganisms isolated from environmental water by quantifying the reproducibility of synthetic gold nanoparticles and the ability of large-amount synthesis.
The manuscript is well structured in general. The analytical methods which are used at the present study are numerous and significant. Nevertheless, the literature should probably be enriched. Twenty-nine references is a limited number for a study which includes all these interesting scientific methods. Moreover, there are few relatively recent references. It would be auspicable if more recent references were added, or at least if the authors would explain the reason for that.
(response)
Thank you for your meaningful comments and suggestions. We improved and modified our manuscript according to your advice such as adding new relatively recent references ([9-11]) to Introduction.
Another important issue is that in the Introduction part the authors do not present clearly the objectives of the present study. They may describe what was carried out but, unfortunately, they do not make clear what was the need of fulfilling this study as well as which are its novelty aspects. This is important for every scientific paper, so please consider and revise.
(response)
Thank you for your advice. Our purpose is to identify the microorganisms that have strong resistance against heavy metals and accumulate the heavy metals around the cell body. We used the color change of gold nanoparticles synthesis for the screening of bacterial strains. This new concept of screening can identify the novel bacterial strains for bioremediation. We added these sentences to Introduction.
The “Results and Discussion” part could be improved with more figures, tables or the presentation of data in diagrams in order the manuscript becomes more interesting to the reader. The “Conclusions” part should definitely be enhanced so that it meets the requirements for publication in such a significant international journal like “Materials”.
(response)
Thank you for your advice. We have added a schematic diagram of the process of reducing gold ions to gold particles by bacteria. And we have supplemented and enhanced the content of the conclusion in the manuscript.
As for the English language style, although I do not feel qualified to judge about it probably requires some editing and needs to be improved throughout the manuscript.
(response)
Thank you for your comment. After revision process, we will submit our manuscript to English editing service.
For the above reasons major revision should be carried out before publication.
(response)
Thank you for your comment. We have adjusted and moved some of the figures, added necessary comparison data, and revised some descriptions and wording.
Some comments and suggestions for the authors concern the abstract, at first. The chemical elements are there referred by their full names while at the rest of the manuscript they are listed by their symbols which is more often used. Please consider and revise so that all elements are listed in a uniform manner throughout the whole text.
(response)
Thank you for your correction. We have unified the expression form of the chemical elements in the manuscript.
Figure 1:
The authors could provide more information regarding the study area, in order to make it more clear to the reader where exactly it is located. Apart from the images that are already depicted, probably an image of the study area’s position in Japan would help.
(response)
Thank you for your advice. We have added the name information of the sampling spots to the figure legend, as well as the prefecture and city in Japan where the sampling spots are located.
Materials and Methods:
At this part the authors describe the methodologies that they followed for the accomplishment of the present study. Unfortunately they do not mention the researchers by which these methodologies were applied for the first time. Meaning, according to whom researchers are these methodologies applied? No references are mentioned following each methodology. Unless they are applied here for the first time. If this is the reason, please state and clarify that. If not, please add the corresponding references.
(response)
Thank you for your comment. The method of screening strains in this study using digitizing the color image is applied here for the first time. When counting and calculating the average particle size of the gold nanoparticles synthesized by K-142, the ImageJ method was used, which has been indicated in the manuscript (reference [13]). We have added the reference of SPR (reference [12]), the reference of the methods and techniques of XRD measurement (reference [14]) and the information about AMCSD (American Mineralogist Crystal Structure Database) (reference [24]). In addition, we have showed the reference about LIVE/DEAD Biofilm Viability Kit (reference [15, 16]). We hope that by supplementing and marking references, the article could become more evidence-based and persuasive.
Figure 5:
The vertical axe “y” of the diagram presents the comprehensive evaluation (total value). What is the unit of this value? The unit by which a value is evaluated must always be mentioned. Please clarify that. The same should probably be considered for the “amount” in the columns. In what way these parameters are quantified? Please consider an make that clearer to the reader.
(response)
Thank you for your comment. The value has no unit and it may just digitize the color change of the samples after the addition of auric acid for 48 hours. The digitizing method or what we call the “examples” of the digitization were showed in the supplement data. The samples which turned dark purple were recorded as “1”, light purple or other colors were recorded as “0.5”, and the ones with no color change were recorded as “0”. This rapid screening method can identify the heavy metal resistance strains from the library. We added the word of “a.u.” at the vertical axis in the figure. ( The new serial number of the corresponding figure is Figure 1)
Figure 8b:
Please add the corresponding reference by which the presented parameters (wavelength, absorbance) were derived. Respective bibliographic references should probably be added for Figures 9, 10, 15, 16, as well.
(response)
Thank you for your comment. It is known that the SPR (Surface Plasmon Resonance) absorption peak of gold nanoparticles is around 550 nm, and the specific shape changes with the shape and size of the gold nanoparticles. The ring patterns were already shown in Figure 9 and 15, which were in the upper right position of the result image. The gold crystal standard in Figure 10 obtained from AMCSD (American Mineralogist Crystal Structure Database). The fluorescent staining results in Figure 16 are analyzed with reference to the instructions of the staining reagents “FilmTracerTM LIVE/DEAD Biofilm Viability Kit”. We added the references about these methods to Materials and Methods just like the response to the previous comment.
Conclusions:
As mentioned above, this part needs definitely to be improved.
(response)
Thank you for your advice. We have supplemented and enhanced the content of the conclusion in the manuscript according to your meaningful advice and comments.
Reviewer 3 Report
This paper reported the microorganisms isolated from environmental water, which was investigated and screened to quantify the reproducibility of synthetic gold nanoparticles and the ability of large-amount synthesis. The results showed that the microorganisms of the genus Bacillus (K-142) showed good activity, which can synthesize gold, cadmium, lead nanoparticles, and the deposition efficiency of silver, cadmium, lead, copper and aluminum reached 64.8%-99.2%. In addition, K-142 has good survival ability and has the potential to be used in the concentration and recovery of heavy metals in environmental water. Therefore, the manuscript is suitable for Materials. However, some revisions should be made by taking into account the followings.
(1) The results show that K-142 has better survival ability. I think the author's research on the life span of K-142 is of great significance.
(2) The author used K-142 to transform several metals. It is necessary to study the appropriate concentration of K-142, the rate and efficiency of metal conversion.
(3) We know that there are various metals in environmental water, and it is recommended that the author could conduct transformation studies that contain more than two metal ions at the same time when conditions permit.
Author Response
Reviwewer3
Comments and Suggestions for Authors
This paper reported the microorganisms isolated from environmental water, which was investigated and screened to quantify the reproducibility of synthetic gold nanoparticles and the ability of large-amount synthesis. The results showed that the microorganisms of the genus Bacillus (K-142) showed good activity, which can synthesize gold, cadmium, lead nanoparticles, and the deposition efficiency of silver, cadmium, lead, copper and aluminum reached 64.8%-99.2%. In addition, K-142 has good survival ability and has the potential to be used in the concentration and recovery of heavy metals in environmental water. Therefore, the manuscript is suitable for Materials. However, some revisions should be made by taking into account the followings.
(response)
Thank you for your meaningful comments and suggestions. We improved and modified our manuscript according to your advice.
(1) The results show that K-142 has better survival ability. I think the author's research on the life span of K-142 is of great significance.
(response)
Thank you for your comment. Actually, in order to describe the viability of K-142, we have once tried to draw the proliferation curve of K-142. In the long-term culture, we found that the colony of K-142 is very viscous and that cause K-142 not easy to separate well from liquids such as water and liquid culture. Therefore, we do not give data on the proliferation curve. However, in the daily culture process, we found that the colony of K-142 is larger and thicker than other strains in the same cultivation time, and the colony of K-142 can be stored at 4 degrees for about half a year but still synthesize Au nanoparticles normally. Although there is no systematic and intuitive scientific data to show, we think that the above-mentioned phenomena could reflect K-142's stronger viability than other strains in the screening library to a certain extent. We added these explanations to Result.
(2) The author used K-142 to transform several metals. It is necessary to study the appropriate concentration of K-142, the rate and efficiency of metal conversion.
(response)
Thank you for your advice. I agree with your opinion. It is necessary to study the appropriate concentration of K-142 for the rate and efficiency of metal conversion. Because of the viscosity and aggregation of K-142, it is difficult to spread the cells of K-142 in the solution. This viscosity may affect the resistance of heavy metals and accumulation of metals around the cells. However, the handling of K-142 with high viscosity increased the difficulty to know the appropriate concentration for metal accumulation. From these reason and limit of revision due (10 days), we can not show the appropriate concentration of K-142 in this paper. We plan to make the other ideas to quantify the amount of K-142 in the future works.
(3) We know that there are various metals in environmental water, and it is recommended that the author could conduct transformation studies that contain more than two metal ions at the same time when conditions permit.
(response)
Thank you for your suggestion. It will be interesting and meaningful data of studying more than two metal ions at the same time. This part of the study can also make the idea of applying K-142 to the environment more convincing. However, this part of experiment may have required a lot of operations, which is difficult to complete it in a short time of 10 days (The limit of revision is only 10 days). We are glad to be inspired by your suggestion and we are planning to design related experiments and conduct a detail study about it in the future publication.
Round 2
Reviewer 1 Report
The paper is much improved after moficiation. It is fine for me to publish.
Reviewer 2 Report
The revised manuscript with the title "Metal accumulation using a bacterium (K-142) identified from environmental microorganisms by the screening of Au nanoparticles synthesis." as well as the cover letter with the authors' responses were carefully checked. The manuscript presents significant improvement after the first round of reviews. Most of the reviewers' comments were addresed, hence the manuscript can be accepted in the present form.